# Circulating miR-451a Expression May Predict Recurrence in Atrial Fibrillation Patients after Catheter Pulmonary Vein Ablation

**DOI:** 10.3390/cells12040638

**Published:** 2023-02-16

**Authors:** Ricardo Lage, María Cebro-Márquez, Marta E. Vilar-Sánchez, Laila González-Melchor, Javier García-Seara, José Luis Martínez-Sande, Xesús Alberte Fernández-López, Alana Aragón-Herrera, María Amparo Martínez-Monzonís, José Ramón González-Juanatey, Moisés Rodríguez-Mañero, Isabel Moscoso

**Affiliations:** 1Cardiology Group, Centre for Research in Molecular Medicine and Chronic Diseases (CIMUS), Universidade de Santiago de Compostela, 15782 Santiago de Compostela, Spain; 2Department of Cardiology and Coronary Unit and Cellular and Molecular Cardiology Research Unit, Institute of Biomedical Research (IDIS-SERGAS), University Clinical Hospital, 15706 Santiago de Compostela, Spain; 3Centro de Investigación Biomédica en Red de Enfermedades Cardiovasculares (CIBERCV), 28029 Madrid, Spain

**Keywords:** atrial fibrillation, microRNAs, biomarker, recurrence, miR-451a, scar percentage

## Abstract

Atrial fibrillation is the most prevalent tachyarrhythmia in clinical practice, with very high cardiovascular morbidity and mortality with a high-cost impact in health systems. Currently, it is one of the main causes of stroke and subsequent heart failure and sudden death. miRNAs mediate in several processes involved in cardiovascular disease, including fibrosis and electrical and structural remodeling. Several studies suggest a key role of miRNAs in the course and maintenance of atrial fibrillation. In our study, we aimed to identify the differential expression of circulating miRNAs and their predictive value as biomarkers of recurrence in atrial fibrillation patients undergoing catheter pulmonary vein ablation. To this effect, 42 atrial fibrillation patients were recruited for catheter ablation. We measured the expression of 84 miRNAs in non-recurrent and recurrent groups (45.2%), both in plasma from peripheral and left atrium blood. Expression analysis showed that miRNA-451a is downregulated in recurrent patients. Receiver operating characteristic curve analysis showed that miR-451a in left atrium plasma could predict atrial fibrillation recurrence after pulmonary vein isolation. In addition, atrial fibrillation recurrence is positively associated with the increment of scar percentage. Our data suggest that miRNA-451a expression plays an important role in AF recurrence by controlling fibrosis and progression.

## 1. Introduction

Atrial fibrillation (AF) is the most common sustained arrhythmia and raises the risk of ischemic stroke and death 5-fold and 2-fold, respectively, increasing long-term incapacity, which has a great impact on the patients’ health and the economic health system of developed countries [1]. Among patients in the Framingham Heart Study population, 37% developed AF after 55 years of age in those who reached that age [2]. AF has a multifactorial and partially known etiology that is closely related to multiple risk factors for cardiovascular disease (CVD), including being elderly, a male gender, hyperthyroidism, hypertension, diabetes, and heart failure, which disturb normal atrial electrophysiology [3]. AF patients suffer a worsening in quality of life, increasing the incidence of ischemic stroke and heart failure, having a high impact on the mortality rate compared to the general population [4]. The pathophysiology of AF involves a complex relationship between the comorbidities, previous state of myocardium, structural and electrical alterations, and myocardial oxidative and/or proinflammatory stress [5,6,7,8,9] associated with an increased risk of mortality in patients with cardiovascular comorbidities [10,11,12]. Although there has been significant progress over the past decade in the understanding of mechanisms of AF and its treatment, the current management of patients with AF is limited to rhythm control in order to reduce symptoms and prevent acute complications, and morbi-mortality associated with AF still remains extremely high. Accordingly, AF frequently recurs after ablation treatment [13].

MicroRNAs (miRNAs) are small non-coding RNAs, comprising of 18–25 nt, that regulate the transcription of their targeted mRNAs. In the last years, miRNAs have been involved in the pathophysiology of different CVDs, such as acute coronary syndrome, coronary artery disease, heart failure, and arrhythmias [14,15,16]. Different studies demonstrated selective changes in the miRNA profile (tissue-specific and blood) in pathophysiologic processes associated with CVD, both in human samples and animal models [14,15,16,17,18]. Furthermore, miRNAs play a central role in stress responses, such as electrical and structural remodeling, fibrosis, vascular inflammation, and atherosclerosis by suppressing gene expression, all these mechanisms being directly related to AF development and maintenance [19,20]. Current data clearly suggest that miRNAs have the potential to contribute to and characterize the pathophysiological status of patients with AF [21]. Previous reports have highlighted miRNAs as biomarkers or as therapeutic targets in AF patients or in vivo and in vitro models [22,23,24,25]. Indeed, the high stability of circulating miRNAs confers them potential for being used as prognostic and diagnostic biomarkers of CVD [20,26,27]. Moreover, microRNAs show their involvement in several arrhythmias in models of human cardiomyocytes derived from induced pluripotent stem cells [28,29,30]. Several miRNAs have been described as possible biomarkers of AF, such as miR-1, miR-328, miR-21, miR-26, miR-29, miR-34, or miR-133 [31]. However, a greater number of clinical studies are needed to prove its usefulness compared to conventional biomarkers. Furthermore, it has been described that there are differences in transcardiac miRNA expression levels that determine which miRNAs are released or absorbed by the myocardium [32,33,34]. In this sense, Soeki et al. found that the local production of miR-328 in the left atrium may be involved in the process of atrial remodeling in patients with AF [35]. In this study, we aim to identify a microRNA profile associated with AF recurrence after catheter ablation. Finally, we also investigated if these identified miRNAs might be implicated in mechanisms that contribute to AF recurrence.

## 2. Materials and Methods

### 2.1. Subjects

We have included 42 consecutive patients with paroxysmal and persistent and persistent long-term AF subjected to pulmonary vein ablation at the University Clinical Hospital of Santiago de Compostela. The exclusion criteria were an age under 18 years, pregnancy, and any latent infectious condition; no history of chronic kidney disease, osteoarthritis, or malignancy were present. Patient characteristics, such as age, gender, medical history, cardiac function, and ECG findings, were recorded during follow-up. The study complies with the Declaration of Helsinki and was approved by the Clinical Research Ethics Committee of Galicia (MRM-miRAF-2017-01). All of the patients signed informed consent forms.

### 2.2. AF Assessment

AF is recorded using a 12-lead electrocardiogram (ECG) before ablation within a period of 6 months. Routinely, a magnetic resonance imaging (MRI) or computed tomography (CT) was performed and used to guide the manipulation of the catheter at the time of the procedure.

### 2.3. Description of the Surgery Intervention and Sample Collection

As described by López-Canoa et al. [36,37], patients were submitted to a night of fasting. First, before the ablation procedure, peripheral blood was obtained using an 18-G butterfly cannula with a two-syringe technique from an ante-cubital vein; the first 5 mL were discarded and the second 5 mL were collected. Blood from the left atrium was obtained through the transeptal sheath, immediately after the transeptal puncture and previous to heparin administration. EDTA-tubes were used to collect blood samples and were immediately centrifuged to collect plasma that was stored at −80 °C for subsequent analysis.

### 2.4. Ablation Procedure, Definition of Scar Percentage and Patient Follow-Up

As previously reported by López-Canoa et al. [36,37], point-by-point radiofrequency catheter ablation was performed in all patients using contact force sensing technology (SmartTouch, Biosense Inc.). Ipsilateral pulmonary vein isolation was the endpoint; 24–36 h after the procedure, the majority of the patients were discharged. During AF, the left atrium may enlarge and stretch to accommodate the elevated pressure and volume, which can cause scarring and injury to the atrium. The scar percentage is defined by the size of the left atrial fibrotic area, derived from the voltage map created using the spiral catheter and a 3-dimensional mapping system (Carto-3, Biosense Webster, or NavX velocity) with the acquisition of at least 1000 data points per atrium. For at least 3 months, oral anticoagulation was maintained and, after this period, continued lifelong in those patients with a CHA2DS2-VASc score of ≥1 in men or ≥2 in women. During the blanking period (3 months), it is the standard of care at our institution to continue antiarrhythmic drug therapy (ADT). In non-recurrent patients after these 3 months, determined by a 24 h Holter recording and clinical evaluation, ADT was discontinued and only restarted in case of relapse. After a second recurrence, outside of the blanking period and after ADT resumption or electrical cardioversion if needed, patients are referred for another ablation procedure. The patients’ follow ups were performed at 3, 6, 12, and 24 months after the index procedure medical examinations in which the detailed history, physical examination, and 12-lead electrocardiogram were registered. AF documented by electrocardiography or Holter, atrial flutter, or atrial tachycardia >30 s in duration and coming about outside the blanking period were considered AF recurrence.

### 2.5. RNA Extraction and miRNA Quantification

Sequences of 84 different predesigned mature miRNAs (listed in Appendix A) were detected using a Human Cardiovascular Disease miScript miRNA PCR Array (MIHS-113Z, Qiagen, Hilden, Germany), as previously described [38]. First, a miRNeasy Serum/Plasma Advanced Kit (Qiagen, Hilden, Germany) was used to isolate total RNA. A miScript II RT kit (Qiagen, Hilden, Germany) was used to obtain cDNA in a SimpliAmp Thermal Cycler (Applied Biosystems, Carlsbad, CA, USA). Afterwards, cDNA was preamplificated with a miScript PreAMP PCR Kit (Qiagen, Hilden, Germany) using a miScript PreAmp Universal Primer and Human Cardiovascular Disease miScript PreAmp Pathway Primer Mix (MBHS-113Z, Qiagen, Hilden, Germany). Exogenous spike-in control cel-miR-39-3p was added to the reaction to evaluate extraction efficiency. RT-qPCR was performed using the miScript SYBR Green PCR kit (Qiagen, Hilden, Germany) in a QuantStudio™ 7 Flex Real-Time PCR System, 384-well plates (Applied-Biosystems, Carlsbad, CA, USA), according to the manufacturer’s indications. The miScript miRNA PCR Array Data Analysis Tool (Qiagen, Hilden, Germany) was used for all calculations. Briefly, only miRNAs with Ct values <30 in all samples were considered for subsequent analysis. miRNAs normalized expressions are represented by ΔCt, calculated by subtracting the global geometric mean signal from individual miRNA Ct values. The 2^−ΔΔCt^ method was used to calculate miRNAs fold change.

### 2.6. MicroRNA Pathway Analysis and Target Prediction

miRNA target genes were retrieved by the miRWalk 3.0 database (http://mirwalk.umm.uni-heidelberg.de/ (accessed on 15 May 2022)). EnrichR was used for GO terms and KEGG pathway enrichment analyses (https://maayanlab.cloud/Enrichr/(accessed on 15 May 2022)). MiRNA-genes-pathways networks were visualized with Cytoscape software (http://cytoscape.org/ (accessed on 28 June 2022)). In silico analyses were performed to altogether elucidate the functional role of differentially regulated miRNAs.

### 2.7. Statistical Analysis

The Shapiro–Wilk test was performed to test the normality of distribution. The Mann–Whitney test, Fisher’s test, and ANOVA, followed by Tukey’s post hoc test, Spearman’s correlation, and the area under curve (AUC) receiver operating characteristic curve analysis were performed with GraphPad Prism 9 (GraphPad Software Inc., San Diego, CA, USA). Multivariate logistic regression analysis was performed to investigate the association between miRNA levels, recurrence, and scar percentage; analysis was performed with IBM SPSS Statistics version 23.0 for Windows (IBM Corp., Armonk, NY, USA). The results of the logistic regression analysis are reported as an odds ratio (OR) and 95% confidence interval (CI). In all analyses, a two-tailed *p* < 0.05 was considered to be significant.

## 3. Results

### 3.1. Baseline Characteristics

The demographic, clinical, and treatment characteristics of the patients are shown in Table 1. A total of 42 AF patients were included in the study. Patients had a mean age of 59.4 ± 9.1 years, 69% were male, and their body mass index (BMI) was 29.3 ± 5 kg/m^2^; 11.9% of the participants had a history of diabetes, and 54.8% arterial hypertension. In total, 84% of the patients remained free of recurrence at 12 months of follow-up. There was no significant difference in age, gender, BMI, diabetes, hypertension, and smoking between patients with or without recurrence. However, there were significant differences in scar percentage and AF pattern. In total, 45.2% of patients recurred during follow-up (Table 1).

### 3.2. Differential Expression of microRNAs in No-Recurrence vs. Recurrence Patients

Plasma expression profiles for a panel of 84 miRNAs showed that miRNAs levels are differentially regulated according to AF recurrence in both peripheral and atrial blood. Thus, compared to the non-recurrence group, in peripheral blood, hsa-let-7b-5p expression is decreased in recurrence patients; whereas hsa-miR-328-3p expression is increased in those with AF recurrence (Figure 1A and Appendix A). In left atrial blood, hsa-miR-451a and hsa-miR-486-5p expressions are decreased in the recurrence group; however, hsa-miR-328-3p expression is increased in recurrence patients (Figure 1B and Appendix A).

Area under curve data show that only left atrial levels of hsa-miR-451 are a good predictor of recurrence (Figure 2A,B).

### 3.3. Association between miR-451a Expression, Scar Percentage and AF Recurrence

According to previous data, atrial fibrosis was a powerful and independent predictor of arrhythmia recurrence [39]. In our study population we found an association between scar percentage and AF recurrence (Figure 3A,B).

To further evaluate the prognostic value of the circulating miR-451, binary logistic regression analyses were performed, including scar percentage and AF recurrence. After adjustment, odds ratios were 0.715 and 1.036, respectively, 95% Cis were 0.56–0.91, *p* = 0.007 and 1.001–1.072, *p* = 0.041. An analysis of the sensitivity and the specificity of plasma miR-451 expression of the corresponding receiver operating characteristic curve to discriminate patients who subsequently had AF recurrence with a higher scar percentage show an AUC of 0.817 (95% CI: 0.69–0.945) with a *p* = 0.0004 (Figure 4 and Appendix A).

### 3.4. KEGG Pathways and Prediction Targets

KEGG signaling pathways and gene targets were further investigated using in silico biological analysis. Enrichment analysis of the known gene targets of miR-451a showed several KEGG pathways including a Ca^2+^ signaling pathway, TGF-beta signaling pathway, mTOR signaling pathway, TNF signaling pathway, and estrogen signaling pathway (Figure 5).

## 4. Discussion

The present study aims to perform a miRNA expression analysis to elucidate a clinical biomarker in AF patients referred for pulmonary vein ablation and the impact on clinical outcomes. The main finding of the study is that miR-451a expression in the left atrium is significantly downregulated in patients with AF recurrence, as compared to patients without recurrence. From the clinical standpoint, scar percentage and AF pattern is elevated in patients with recurrence. Receiver operating characteristic curve analysis ascertained that the variant expression of miR-451a in the plasma of patients with AF could predict AF recurrence after pulmonary vein isolation. Moreover, AF recurrence is positively associated with the area size of the scar in the left atrium. Other studies have identified different microRNA profiles in AF development and progression [40,41,42,43] and also AF recurrence after cardioversion [44].

### 4.1. Background

The association between CVDs and miR-451a has been previously reported, being a reliable biomarker for AMI diagnosis [45], atherosclerosis [46], dilated cardiomyopathy [47], or heart failure [32]. It also has been described that it plays a role in diabetic cardiomyopathy in the murine obese model [48], and in cardiac hypertrophy, both in human or animal models [49]. The miR-451 level was found to be decreased in human cardiac hypertrophy and subsequent sudden cardiac death, regulating cardiac hypertrophy and cardiac autophagy by targeting TSC1, showing a negative correlation with left ventricular mass and becoming a potential therapeutic target for the disease [49,50]. Thereby, miR-451a is one of the most downregulated miRNAs in a murine model of cardiac hypertrophy induced by transverse aortic constriction surgery [51]. In contrast, it was reported that cardiac miR-451 expression is increased as a consequence of ischemic preconditioning or a high-fat diet [52]. Deng et al. also reported that miR-451a regulates cardiac fibrosis and angiotensin II-induced inflammation by targeting TBX1, becoming a possible therapeutic target for treating cardiac hypertension [53]. In addition, it was reported that miR-451 inhibits high mobility group box 1 (HMGB1), increasing the cardiomyocyte protection against hypoxia/reoxygenation injury [54], and also protects cardiomyocytes against ischemia/reperfusion-induced death by targeting the CUG triplet repeat-binding protein2 (CUGBP2)-cyclooxygenase-2 (COX-2) pathway [55]. Additionally, miR-451a has been reported to be used as an early biomarker of CAD in patients with plaque rupture [56].

### 4.2. AF Patients

Previous results showed that changes in miRNA profile have been associated with the development and maintenance of AF. Many miRNAs are proved to be regulating the atrial-specific ion channel TASK-1, upregulated in AF [57]. A decreased miR-1 expression in persistent AF patients is associated with an increase in potassium currents (I_K1_) [58]. Although miR-1 is also associated with the regulation of Ca^+2^ channels, there are no studies that demonstrate its involvement in the pathophysiology of AF [59]. Otherwise, the decreased myocardial expression of miR-26 and increased expression of miR-328 in AF patients and canine models are accompanied by a significant increase in I_K1_ currents, Kir2.1 expression, and a significant decrease through L-type Ca^+2^ channels (ICaL), respectively [60,61]. The restoration of normal levels of these miRNAs balances ionic currents. Increased myocardial expression of miR-499 might also participate in AF-associated electrical remodeling by altering Ca^+2^-dependent potassium currents [62,63]. Otherwise, different miRNAs have been identified as potential regulators of fibrosis, the central process of structural remodeling in AF [26]. Altered miR-21 expression is the most consistent change; AF patients show an increase in atrial expression of miR-21 [64]. miR-21 silencing prevents atrial fibrosis, a substrate for AF, in murine models of myocardial infarction [65]. It has also been described that circulating miR-21 correlates with left atrial low voltage areas, and that is associated with procedure outcomes in patients with persistent AF undergoing ablation [22]. Decreased miR-26 and miR-29 atrial expression has been described in canine models of AF and heart failure, respectively. miR-29 expression correlates inversely with extracellular matrix protein levels and the development of AF [66]. Decreased circulating and atrial levels of miR-29 have also been observed in FA patients [66]. Additionally, increased levels of TGF-β and its receptor, known key profibrogenic factors, seems to be related to decreased miR-133 and miR-590 expression in canine models of AF [67].

In the setting of patients with AF referred for pulmonary vein isolation, we found significant differences in the expression of let-7b-5p, miR-328-3p, miR-486-5p, and miR-451a that are differently regulated in peripheral blood, as compared to locally in the left atrium, although let-7b-5p [68], miR-328-3p [35,69], and miR-486-5p [68,70] were previously reported to be implicated in AF, but only miR-451 seems to be a good predictor of AF recurrence based on AUC analysis, considering the AUC value between 0.7–0.9 as a moderate prognostic value. Since the atrium and surrounding tissues secrete inflammatory cytokines and matrix metalloproteinases, which attract macrophages and neutrophils and promote remodeling of the fibrosis in the atrial myocardium that causes AF, levels of microRNAs and other biomarkers in left atrial blood may be different from those found in the peripheral blood [71]. Several studies have demonstrated that atria structural remodeling, related to cardiomyocyte apoptosis, inflammation, and the activation of fibrotic pathways, is associated with AF pathogenesis [72,73]. Although Zhang et al. demonstrated that miR-155 levels after 12-month follow-up predict the recurrence of AF after cardioversion in plasma, no previous results showed the regulation of miR-451a expression in AF patients in left atrium blood before the ablation procedure. Deng et al. [53] described how miR-451a is directly related to the attenuation of fibrosis induced by angiotensin II. Furthermore, Scrimgeour et al. [51] also reported that after pathological stimulation, miR-451a prevents the increased expression of MMP-9 and MMP-2 in cardiomyocytes. Additionally, an increase in MMP-2 and MMP-9 produces extracellular matrix degradation, reduces the density of transverse tubules, and alters the management of Ca^2+^ in the myocardium of heart failure patients. Accordingly, it has been reported that MMPs are increased in AF patients [74,75,76,77]. Therefore, the downregulated expression of miR-451a in patients with recurrence could be due to an increase in atrial myocardial fibrosis [78].

In silico analysis of validated target genes and signaling pathways suggest that miR-451a downregulation could predict AF recurrence [76]. Ebana et al. indicated that AF pathogenesis is also related to the mTOR pathway, although the specific mechanisms are still unclear [79]. Previous reports show how AMPK is activated in atrial tissues and atrial cardiomyocytes from dogs with AF. In addition, AMPK activation in the myocardium of the atrium has also been demonstrated in AF patients [80]. Li J et al. demonstrated that, in a cardiac injury model, miR-451 inhibition increases Ca^2+^ binding protein 39 (CAB39) expression, and the AMPK*α* signaling pathway increased the activation, both in vivo and in vitro [52]. Furthermore, CAB39 and its binding partners LKB1 (liver kinase-B1) and STRAD (LYK5) are downregulated in human atrial biopsies in patients with paroxysmal AF. The Ca^2+^ signaling pathway is also related to AF; several reports suggested that abnormal intracellular Ca^2+^ handling contributes to AF progression in a spontaneous AF in a murine model [81]. AF patients show higher levels of the miR-451a targeted genes MIF and MYC. *MIF* is a pleiotropic inflammatory cytokine, for which the mechanism of action is related to AF progression, promoting fibrosis development through the activation of the TGF-β signaling pathway, and was described to be affected by miR-451a downregulation. Pan et al. postulated that, based on the link between genes, pathways, and AF, MYC participates in AF pathogenesis [78]. Other signaling pathways are also implicated in AF development and progression. Signaling pathways, such as Ang II/JAK/STAT3 [82], NF-κB [83], TNF-α [84], PI3K/Akt [85], and Wnt [86], are involved in key processes for the proper maintenance of cardiac metabolism and for the development and progression of AF, such as atrial remodeling, proinflammatory state, and fibrosis [87].

Th scar percentage is higher in recurrence patients, which is in accordance with in silico predicted KEGG signaling pathways in which miR-451a is implicated. These results are in accordance with previous data that show that baseline fibrosis is a risk factor for AF recurrence [88].

### 4.3. Limitations

The present study has some limitations that needs to be taken into account at the time of data interpretation. First and very important, the study was conducted with a small sample size in a single center. Thus, the expression of miR-451a in AF patients needs to be validated in a larger sample size. One of the limitations of our study is that it is a non-randomized retrospective study. The lack of statistical power could be the cause of the lack of association with different clinical variables. It should be noted that our objective was to analyze the link between miRNA expression with clinical variables in patients with AF undergoing ablation, with low voltage areas (scar, in our data) in electroanatomic voltage mapping. Only patients whose point-by-point radiofrequency ablation was performed with contact force sensing technology were included; this avoids the bias of recurrence due to patient-specific procedures. In our study, magnetic resonance imaging was not performed; it is a non-invasive and very precise technique for evaluating atrial myopathy; this would have provided correlation data with miRNAs and the scar percentage. Furthermore, 16% of our patients had already undergone pulmonary vein ablation. Patients with radiofrequency lesions beyond the pulmonary veins were not included in the analysis because they may interfere with the level of myopathy determined by electroanatomic voltage mapping. Thus, the quantification of the scar percentage was performed outside the pulmonary veins, so this does not influence the conclusions of the study. To conclude, the temporal pattern and intermittent ECG monitoring determined the AF burden, although this did not really correspond to long-term ECG monitoring.

## 5. Conclusions

miRNA-451a expression might play a critical role in AF recurrence by controlling fibrosis and progression. However, more experimental studies are needed to stablish miR-451a as a biomarker of AF recurrence in catheter pulmonary vein ablation treated patients before its use as a biomarker or therapeutic target. Further, perspective studies with larger sample sizes are required to assess the potential of miR-415a as a biomarker for predicting recurrence in patients with AF.

## Figures and Tables

**Figure 1 cells-12-00638-f001:**
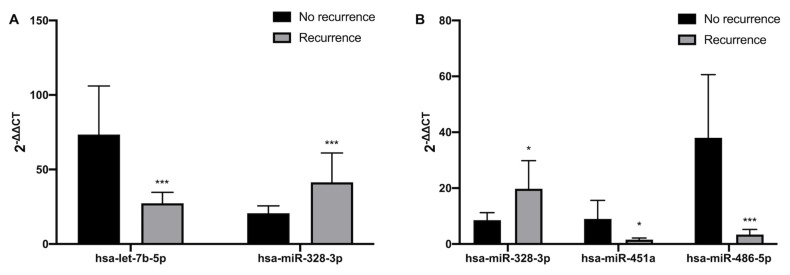
Differential expression of miRNAs in no-recurrence vs. recurrence AF patients (**A**) plasma from peripheral blood (**B**) plasma from left atrium blood. Data are presented as mean  ±  S.E.M. * *p* < 0.05, *** *p* < 0.001.

**Figure 2 cells-12-00638-f002:**
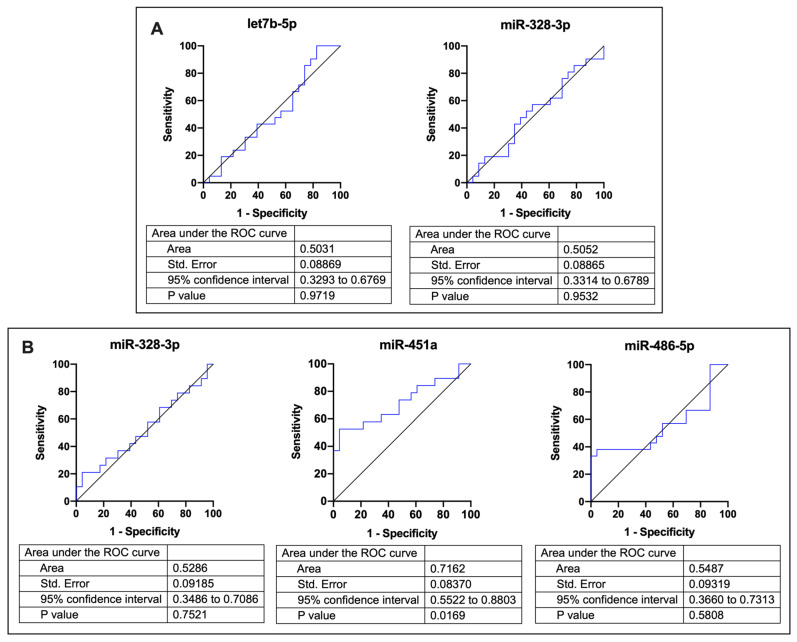
Predictive capacity of AF recurrence. Receiver-operating characteristic curves comparing sensitivity and specificity of (**A**) hsa-let7b and has-miR-328-3p expression in plasma from peripheral blood and (**B**) has-miR-328-3p, hsa-miR-451a, and has-miR-486-5p expression in plasma from left atrial blood, where hsa-miR-451a expression is predicting recurrence in AF patients.

**Figure 3 cells-12-00638-f003:**
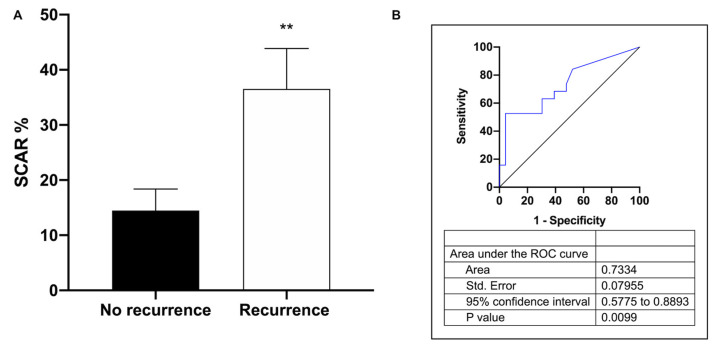
Association between scar percentage and AF recurrence. (**A**) Scar percentage in no recurrence and recurrence patients. (**B**) Receiver-operating characteristic (ROC) curve comparing sensitivity and specificity of recurrence and scar percentage in AF patients. Data are presented as mean ± S.E.M. ** *p* < 0.01.

**Figure 4 cells-12-00638-f004:**
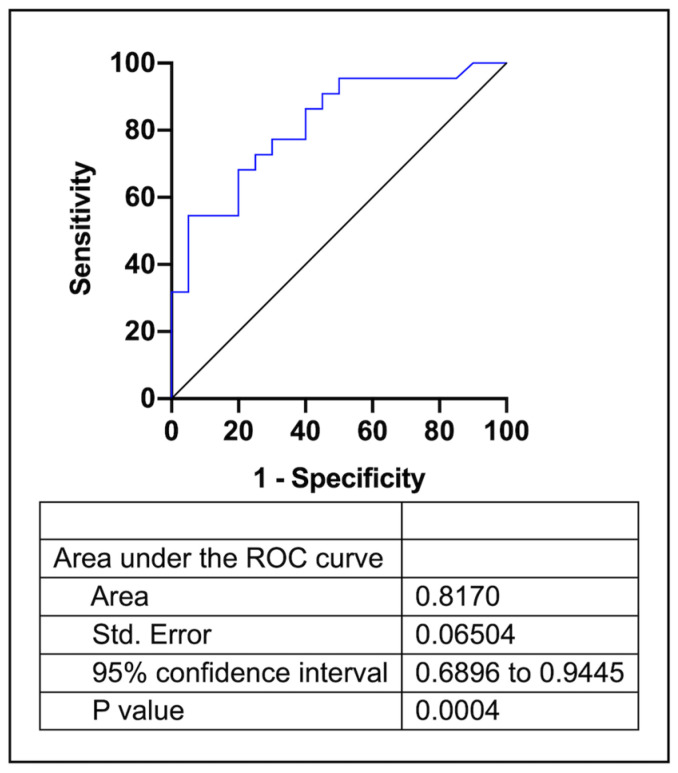
Receiver-operating characteristic curve comparing sensitivity and specificity for the binary logistic regression model.

**Figure 5 cells-12-00638-f005:**
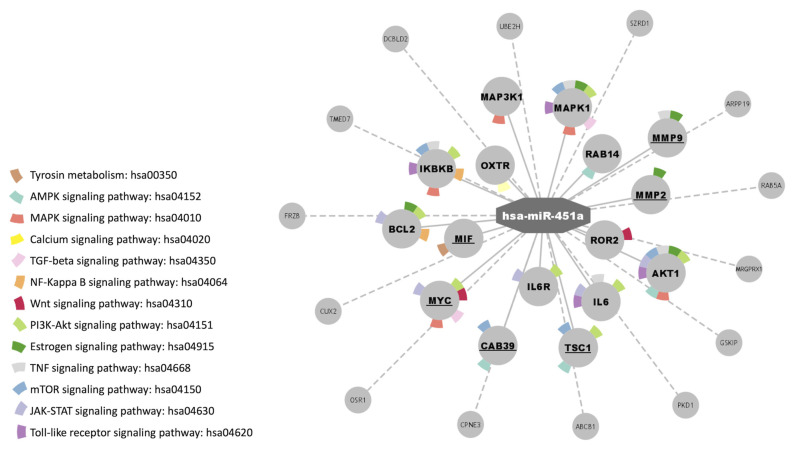
MiRNA-genes-pathways network denoting the relationships between miR-451a, validated target genes (dotted lines), and those related to KEGG pathways (solid lines), created in Cytoscape. Bold genes: belonging to KEGG pathways. Underlined genes: implicated in AF.

**Table 1 cells-12-00638-t001:** Clinical parameters. Data are presented as mean + SD, *n* (%).

Parameter	Total (*n* = 42)	w/o Recurrence (*n* = 23)	Recurrence (*n* = 19)	*p* Value
**Age (years)**	59.4 ± 9.1	57.4 ± 8.8	60.8 ± 8.9	0.078
**Male**	29 (69%)	16 (69.6%)	13 (68.4%)	0.936
**Female**	13 (31%)	7 (30.4%)	6 (31,6%)	0.936
**BMI**	29.3 ± 5	29.7 ± 5.2	29 ± 4.7	0.807
**Pre-existing Conditions**				
**Hypertension**	23 (54.8%)	12 (52.2%)	11 (57.9%)	0.711
**Diabetes**	5 (11.9%)	2 (8.7%)	3 (15.8%)	0.480
**Smoking**	12 (28.6%)	6 (26.1%)	6 (31.6%)	0.695
**Scar %**	24.7 ± 27.9	14.5 ± 18.7	36.4 ± 32.1 ***	0.008
**Tachycardiomyopathy**	8 (19%)	4 (17.4%)	4 (21.1%)	0.763
**Statines**	19 (45.2%)	8 (34.8%)	11 (57.9%)	0.134
**ACEi**	9 (21.4%)	6 (26.1%)	3 (15.8%)	0.418
**ARB**	12 (28.6%)	4 (17.4%)	8 (42.1%)	0.078
**DHP Ca channel blockers**	5 (11.9%)	2 (8.7%)	3 (15.8%)	0.480
**Acenocoumarol**	17 (40.5%)	9 (39.1%)	8 (42.1%)	0.845
**NOAG**	25 (59.5%)	14 (60.9%)	11 (57.9%)	0.845
**Class I ADT**	14 (33.3%)	10 (43.5%)	4 (21.1%)	0.125
**Class II ADT**	32 (76.2%)	17 (53.1%)	15 (46.9%)	0.703
**Class III ADT**	13 (30.9%)	7 (30.4%)	6 (46.2%)	0.936
**Class IV ADT**	4 (9.5%)	1 (4.4%)	3 (15.8%)	0.209
**Cholesterol**	191.5 ± 39.1	195.9 ± 36.4	185.8 ± 56.2	0.711
**LDLc**	112.3 ± 29.3	118.3 ± 30	104.6 ± 33.5	0.276
**HDLc**	54.8 ± 17.8	55.2 ± 19.2	54.4 ± 18.2	0.710
**TG**	119.9 ± 52.5	129.5 ± 60.6	107.6 ± 38.1	0.242
**AF type**				
**Type 1**	12 (28.6%)	11 (47.8%)	1 (5.3%) ***	0.002
**Type 2**	17 (40.5%)	8 (34.8%)	9 (47.4%)	0.903
**Type 3**	13 (30.9%)	4 (17.4%)	9 (47.4%) **	0.013
**Echocardiographic Parameters**				
**LVEF (%)**	59.1 ± 10.6	61.3 ± 8.4	56.2 ± 16.7	0.195
**LA Area**	19.6 ± 6.0	18.7 ± 5.5	20.5 ± 7.7	0.424
**LA Vol**	91.2 ± 46.1	86.8 ± 44.9	96.2 ± 49.8	0.413
**LVEDV**	62.3 ± 33.6	62.8 ± 32	61.8 ± 38	0.944
**LVESV**	25 ± 19	23.5 ± 11.4	27 ± 26.7	0.783
**LVTDD**	40 ± 8.6	40.5 ± 9	39.3 ± 11	0.748
**LVTSD**	28 ± 6.6	28.9 ± 6.9	26.8 ± 8.3	0.376
**EAT Vol**	80.6 ± 52.8	80.5 ± 49.4	80.9 ± 58.7	0.738
**ECG Parameters**				
**HR**	74.2 ± 21.5	72.3 ± 24.3	76.5 ± 21.7	0.136
**PR**	157.7 ± 25	156.9 ± 28.0	160.2 ± 55.8	0.877
**QRS**	95 ± 12.5	93.3 ± 11.1	97.2 ± 25.0	0.480

BMI, body mass index; ACEi, angiotensin-converting enzyme inhibitor; ARB, angiotensin-receptor blocker; DHP Ca channel blockers, Dihydropyridine Ca^2+^ channel blockers; NOAG, new oral anticoagulants; ADT, antiarrhythmic drug therapy; LDLc, low-density lipoprotein cholesterol; HDLc, high-density lipoprotein cholesterol, TG, triglycerides; AF type 1, paroxysmal; AF type 2, persistent; AF type 3, persistent long term; LVEF (%), left ventricular ejection fraction; LA area, left atrium area; LA Vol, left atrium volume; LVEDV, left ventricular end-diastolic volume; LVESV, left ventricular end-systolic volume; LVTDD, left ventricle telediastolic diameter; LVTSD, left ventricle telesystolic diameter; EAT Vol, epicardial fat tissue volume; HR, heart rate; PR, PR interval; QRS, QRS duration. ** *p* < 0.01, *** *p* < 0.001.

## Data Availability

The data presented in this study are available on request from the corresponding authors.

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
