# Peer review of "Circulating miR-451a Expression May Predict Recurrence in Atrial Fibrillation Patients after Catheter Pulmonary Vein Ablation"

_cells, 2023, doi:10.3390/cells12040638_

Round 1

Reviewer 1 Report

Dear Authors,

The manuscript presented to me for evaluation concerns miRNA molecules, which, despite being at the center of interest for many years, still arouse great research interest around the world. Their association with numerous disease entities is getting better and better empirically documented, which at the same time gives the world hope that they will be used as biomarkers of various diseases.

The concept of research is logical and well thought out. The authors measured the expression of 84 miRNAs, although it is a pity that in such a small group (42 patients) and this is a definite limitation of this study.

On the other hand, the examination of 84 miRNAs in such a small group allowed the emergence of miR-451a and the conclusion that miRNA-451a expression might play a critical role in AF recurrence by controlling fibrosis and progression.

I appreciate the criticism shown by the authors towards the manuscript, listing the limitations of the presented studies.

I hope that they will continue this research (the same miRNAs) only on another large group. This could yield tangible results with the development of the biomarker in patients with AF.

Notes to be completed:

1. Methodical:

(a) Line 400: Was UniSp6 spike added to the mixture at the stage of cDNA synthesis to control the correct synthesis and subsequent RT-qPCR reaction?

(b) Line 410-412 "miRNAs normalized expressions are represented ΔCt, calculated by subtracting the global geometric mean signal from individual miRNA Ct values. 2−∆∆Ct method was used to calculate miRNAs fold change.

What miRNAs were used for normalization expression and on what basis they were selected?

(c) How many fold cDNA templates were diluted?

Author Response

Response to Reviewer 1 Comments

Comments and Suggestions for Authors

Manuscript: cells-2183305

Dear Authors,

The manuscript presented to me for evaluation concerns miRNA molecules, which, despite being at the center of interest for many years, still arouse great research interest around the world. Their association with numerous disease entities is getting better and better empirically documented, which at the same time gives the world hope that they will be used as biomarkers of various diseases.

The concept of research is logical and well thought out. The authors measured the expression of 84 miRNAs, although it is a pity that in such a small group (42 patients) and this is a definite limitation of this study.

On the other hand, the examination of 84 miRNAs in such a small group allowed the emergence of miR-451a and the conclusion that miRNA-451a expression might play a critical role in AF recurrence by controlling fibrosis and progression.

I appreciate the criticism shown by the authors towards the manuscript, listing the limitations of the presented studies.

I hope that they will continue this research (the same miRNAs) only on another large group. This could yield tangible results with the development of the biomarker in patients with AF.

We want to thank the Reviewer 1 for his/her valuable comments as well as the positive and constructive comments. We want to apologize to the reviewer for not having clearly explained some parts of the methodology. We have clarified bellow specific methodological commentaries as indicated.

Notes to be completed:

  1. Methodical:

(a) Line 400: Was UniSp6 spike added to the mixture at the stage of cDNA synthesis to control the correct synthesis and subsequent RT-qPCR reaction?

We have added cel-miR39-3p as spike in control to the mixture to assess extraction efficiency. It was also indicated in Materials and Methods section.

(b) Line 410-412 "miRNAs normalized expressions are represented ΔCt, calculated by subtracting the global geometric mean signal from individual miRNA Ct values. 2−∆∆Ct method was used to calculate miRNAs fold change.

What miRNAs were used for normalization expression and on what basis they were selected?

We used all analyzed and reference microRNAs (SNORD61, SNORD68, SNORD72, SNORD95, SNORD96A, RNU6-6P) to calculate the global geometric mean.

(c) How many fold cDNA templates were diluted?

Samples were diluted 20-fold

Reviewer 2 Report

This study measures the abundance of 84 miRNAs in blood samples from peripheral vessels or within the left atrium and shows that downregulation of miR-451a is associated with AF recurrence after PVI. The manuscript contains novel findings.

I will direct a major revision of the revised manuscript for me to check, but it does not require additional experiments.

Major concerns:

It is difficult to see the whole picture of the process when focusing on miR-451a.

(1) Is it possible to show the whole picture with volcano plots?

(2) It would be better to show all bar graphs and ROC curves for let-7-b-5p, miR-328-3p, miR-451a and miR-486-5p. Of these, it can be highlighted that miR-451a is the one that was significant in the ROC analysis.

(3) Please re-analyse the above for peripheral blood and intra-atrial blood respectively.

(4) It would be more directly understandable if you also made a scatter plot of miR-486-5p expression levels and scar size and obtained correlation coefficients.

Specific comments:

Basically, there are too many typos. I cannot check them all by myself. Please revise it carefully yourself, taking my comments into account.

Line 31 – 32: "ROC" and "PVI" should not be abbreviated.

Line 32 – 33: e.g. “The frequency of recurrence correlates with the area size of the scar in the left atrial appendage.”? --- The word 'increment' is not used conventionally, as the size of the scar is a continuous variable.

Line 92: e.g. “Patient characteristics such as age, gender, medical history, cardiac function and ECG findings …”?

Table 1: Add a line for “Female”.

Section 2.2, line 108 – 119: It should be described whether it is increasing or decreasing with reference to the control (No recurrence). Please revise.

Line 120: “The area under curve (AUC)”

Line 135 – 138: “mir-451” should be “miR-451”.

Section 3 (Discussion),

Line 158: “PVA” should not be abbreviated.

Line 163: “PVI” should not be abbreviated.

Line 163 – 164: The word 'increment' is not used conventionally, as the size of the scar is a continuous variable.

Line 164 – 166: This sentence is out of context. Please delete it or put it elsewhere.

Line 169: “Although … studied,” --- Please delete it.

Line 172 – 173: This sentence is out of context. Please delete it.

Line 200: IK1 --- K is capitalized and K1 is subscripted. Please correct for all “Ik1”.

Line 201: Ca2+, calcium --- 2+ is superscripted. Please correct for all “Ca2+” and "calcium"'.

Line 205: L-type Ca2+ channel

Line 239: MMP-2 (metalloproteinase-2)

Line 245 – 248:  “mir-451” should be “miR-451”.

Line 261: “Cab39” should be “CAB39”.

Line 265: “Ca2+” signaling pathway

Section 4.4,

The definition of “scar percentage” is one of the backbones of the paper. It is more helpful to the readers if it is stated emphatically.

Line 368: “4.4. Ablation procedure, definition of scar percentage, and patient follow-up”

Line 373 – 374: e.g. “The scar percentage is defined by the size of the left atrial fibrotic area, derived from the voltage map …”

Author Response

Response to Reviewer 2 Comments

Comments and Suggestions for Authors

Manuscript: cells-2183305

This study measures the abundance of 84 miRNAs in blood samples from peripheral vessels or within the left atrium and shows that downregulation of miR-451a is associated with AF recurrence after PVI. The manuscript contains novel findings.

I will direct a major revision of the revised manuscript for me to check, but it does not require additional experiments.

We would like to first thank reviewer 2 for his/her response to our work and his/her constructive comments. We have addressed the concerns that were raised by editing the manuscript and believe that it will improve significantly as a result of this process. We greatly appreciate the opportunity to submit this revised review manuscript.

Major concerns:

It is difficult to see the whole picture of the process when focusing on miR-451a.

We appreciate the reviewer's suggestions. According to the indications, we have modified or include all suggested figures.

(1) Is it possible to show the whole picture with volcano plots?

We have included volcano plots of peripheral blood and left atrium blood as supplementary Figure S1.

(2) It would be better to show all bar graphs and ROC curves for let-7-b-5p, miR-328-3p, miR-451a and miR-486-5p. Of these, it can be highlighted that miR-451a is the one that was significant in the ROC analysis.

We have included ROC curves of peripheral blood and left atrium blood as Figure 2A and B.

(3) Please re-analyse the above for peripheral blood and intra-atrial blood respectively.

We have included all suggested figures of peripheral blood and left atrium blood.

(4) It would be more directly understandable if you also made a scatter plot of miR-486-5p expression levels and scar size and obtained correlation coefficients.

We have included scatter plots and coefficients of miR-451a expression levels in left atrium blood and scar size/recurrence as supplementary Figure S2.

Specific comments:

Basically, there are too many typos. I cannot check them all by myself. Please revise it carefully yourself, taking my comments into account.

We fully agree with the reviewer, we have revised and changed, as indicated, all specific comments listed below:

  1. Line 31 – 32: "ROC" and "PVI" should not be abbreviated.
  2. Line 32 – 33: e.g. “The frequency of recurrence correlates with the area size of the scar in the left atrial appendage.”? --- The word 'increment' is not used conventionally, as the size of the scar is a continuous variable.
  3. Line 92: e.g. “Patient characteristics such as age, gender, medical history, cardiac function and ECG findings …”?
  4. Table 1: Add a line for “Female”.
  5. Section 2.2, line 108 – 119: It should be described whether it is increasing or decreasing with reference to the control (No recurrence). Please revise.
  6. Line 120: “The area under curve (AUC)”
  7. Line 135 – 138: “mir-451” should be “miR-451”.
  8. Section 3 (Discussion),
  9. Line 158: “PVA” should not be abbreviated.
  10. Line 163: “PVI” should not be abbreviated.
  11. Line 163 – 164: The word 'increment' is not used conventionally, as the size of the scar is a continuous variable.
  12. Line 164 – 166: This sentence is out of context. Please delete it or put it elsewhere.
  13. Line 169: “Although … studied,” --- Please delete it.
  14. Line 172 – 173: This sentence is out of context. Please delete it.
  15. Line 200: IK1 --- K is capitalized and K1 is subscripted. Please correct for all “Ik1”.
  16. Line 201: Ca2+, calcium --- 2+ is superscripted. Please correct for all “Ca2+” and "calcium"'.
  17. Line 205: L-type Ca2+ channel
  18. Line 239: MMP-2 (metalloproteinase-2)
  19. Line 245 – 248: “mir-451” should be “miR-451”.
  20. Line 261: “Cab39” should be “CAB39”.
  21. Line 265: “Ca2+” signaling pathway
  22. Section 2.4,
  23. The definition of “scar percentage” is one of the backbones of the paper. It is more helpful to the readers if it is stated emphatically.
  24. Line 368: “2.4. Ablation procedure, definition of scar percentage, and patient follow-up”
  25. Line 373 – 374: e.g. “The scar percentage is defined by the size of the left atrial fibrotic area, derived from the voltage map …”

Reviewer 3 Report

Comment No. 1: This paper should be edited grammatically.

Comment No. 2: You should add quantitative results to the abstract.

Comment No. 3: The originality of the paper needs to be stated clearly. It is of importance to have sufficient results to justify the novelty of a high-quality journal paper. The Introduction should make a compelling case for why the study is useful along with a clear statement of its novelty or originality by providing relevant information and providing answers to basic questions such as: What is already known in the open literature? What is missing (i.e., research gaps)? What needs to be done, why and how? Clear statements of the novelty of the work should also appear briefly in the Abstract and Conclusions sections.

Comment No. 4: An updated and complete literature review should be conducted and should appear as part of the Introduction, while bearing in mind the work's relevance to this Journal and taking into account the scope and readership of the journal. The results and findings should be compared to and discussed in the context of earlier work in the literature.

Comment No. 5: The literature review is inadequate; you should include relevant publications such as:

10.1177/09544089211069211, 10.1016/j.imu.2017.10.007

Comment No. 6: Result and discussion section can be more improved from the physical point of view.

Author Response

We have carefully read reviewers´comments and we have doubts about Reviewer 3 report, it seems that he/she has sent comments of other paper

The first 4 comments are very general, but 5 and 6 comments do not apply to our paper, comments are about physics, vessels and fluid dynamics.

Comment No. 5: The literature review is inadequate; you should include relevant publications such as:

10.1177/09544089211069211 "Hydrothermal analysis on non-Newtonian nanofluid flow of blood through porous vessels"

10.1016/j.imu.2017.10.007 “Investigating the effect of adding nanoparticles to the blood flow in presence of magnetic field in a porous blood arterial"

Comment No. 6: Result and discussion section can be more improved from the physical point of view.

We wait for the editor's indications to clarify if it is a misunderstanding

Round 2

Reviewer 2 Report

Line 109: "p"ulmonary vein ...

The manuscript much improved. I have no further questions.

Reviewer 3 Report

Now it can be published.